# The Impact of One-Time Relaxation Training on Attention Efficiency Measured by Continuous Performance Test in Depressive Disorders

**DOI:** 10.3390/ijerph19116473

**Published:** 2022-05-26

**Authors:** Kinga Rucka, Monika Talarowska

**Affiliations:** Institute of Psychology, Faculty of Educational Sciences, University of Lodz, ul. Smugowa 10/12, 90-433 Lodz, Poland; monika.talarowska@now.uni.lodz.pl

**Keywords:** depressive disorders, relaxation training, attention efficiency, continuous performance test

## Abstract

**Introduction**: People with depression often complain of dysfunction in cognitive processes, particularly attention. Pharmacotherapy is one of the most commonly used methods of treating depressive disorders and related attention difficulties. Patients also benefit from various forms of psychotherapy and frequently support themselves with alternative therapeutic methods. The purpose of this study was to examine whether a 15-min-long relaxation training session could improve the efficiency of attention and perceptiveness in individuals diagnosed with depressive disorders. **Methods:** Forty-two individuals participated in the study, including 20 individuals diagnosed with recurrent depressive disorder (rDD) and 22 healthy subjects (comparison group, CG). The so-called continuous performance test in the Polish version (Attention and Perceptiveness Test, APT) was applied in the study. In the first stage, the participants completed the 6/9 version of the APT test and then took part in a 15-min relaxation training session (autogenic training developed by the German psychiatrist Johannes Heinrich Schultz). The next step of the study was to perform APT again (parallel version—3/8). **Results:** The analyses showed statistically significant differences (*p* < 0.001) in the results obtained in the two versions of APT between the studied groups (rDD versus CG) in terms of the perceptual speed index. These differences were seen both before and after the introduction of the relaxation training. There was a statistically significant difference in the value of the perceptual speed index before and after the applied relaxation training for the subjects with depression (*p* = 0.004) and for the whole study group (*p* = 0.008). A significant correlation of illness symptom severity with decreased attentional efficiency was observed in the rDD group (perceptual speed index)—both before (*r* = −0.864; *p* < 0.001) and after the relaxation training (*r* = −0.785; *p* < 0.001). **Conclusions:** The continuous performance test (APT) is a reliable indicator of impaired attention efficiency among patients with depressive symptoms compared to healthy subjects. 15-min-long one-time relaxation exercise has a beneficial effect on attention efficiency measured by APT in people with depression.

## 1. Introduction

Cognitive deficits in the course of depressive disorders, predominantly related to memory processes, attention, and the so-called frontal functions, have been deeply explored in recent years [1,2] Cognitive impairment is increasingly regarded as a new and important target of pharmacological treatment [3] and may, in return, contribute to the ineffectiveness of antidepressant therapy and impede full recovery, thus leading to incomplete functional remission [4].

In recent years, there has been a growing interest in the non-pharmacological treatment of psychiatric disorders, with benefits such as improved mental and physical health and better ability to cope with tension and stress [5]. Subjective and objective data support the hypothesis that an integrated central nervous system response underlies the relaxation-induced change in the state of consciousness. For many years, relaxation techniques have been used in the treatment of depression as one way to improve patients’ functioning [6]. Nowadays, such activities are used as an additional form of therapy to traditional treatment methods (pharmacotherapy and psychotherapy) rather than as a stand-alone way to eliminate the symptoms of mental disorders [7].

Jorm et al. [8] observed that the incorporation of relaxation techniques into pharmacological treatment led to a reduction in depressive symptoms. A combination of medication along with relaxation was a more effective method of helping patients than pharmacotherapy alone. However, the visualization-based relaxation techniques used were not as effective in relieving the symptoms of the illness as the psychotherapeutic interventions undertaken along with pharmacotherapy. The cited authors noted that relaxation techniques had great potential in the treatment of depression, especially during the first episode of the illness [8].

A study describing the effects of an eight-week Mindfulness-Based Stress Reduction course on affective symptoms (severity of depression and anxiety) found that mindfulness techniques reduced the tendency to ruminate in individuals with anxiety and depression [9]. Mikicins’ study [10] demonstrated the positive effects of audio-visual relaxation and autogenic training on the generation of *alpha* brain waves (7–12 Hz), which occur in the relaxed state. The dominance of *alpha* waves was also linked to increased creativity and motivation in the examined subjects [11]. The results of the cited studies demonstrated the positive impact of systematic use of audio-visual relaxation as well as autogenic training on both the behavior and well-being of the [10]. In addition, Jacobson’s progressive muscle relaxation is beneficial in reducing symptoms of anxiety and depression. It also results in an increased sense of self-control and an elevated subjective sense of well-being [12].

Ali et al. [13] presented a case report of the treatment of a female patient suffering from a depressive-anxiety disorder, in which—as a consequence of 12 weeks of therapy using relaxation techniques—a significant improvement in the subject’s functioning was demonstrated. The patient scored lower on anxiety and depression scales, and reduced levels of tension, pain, and sleep disturbance were detected [13].

Patients who suffer from depression report numerous subjective complaints of cognitive dysfunction [14]. Depending on the severity of such symptoms, these include attention difficulties or persistence [15]. There are also studies that suggest that deterioration in the efficiency of attention processes had occurred even before the first symptoms of depression appeared [16,17]. It is undeniable that cognitive changes in depression are an important aspect of this disorder [2]. Recent findings suggest that cognitive training can be used to modify and enhance cognitive control in depressed individuals, which in turn improves stress reactivity and emotion regulation. Furthermore, cognitive behavior therapy (CBT) and mindfulness interventions may increase cognitive control, thereby enhancing the ability to regulate negative affect [18].

The purpose of this study was to evaluate the effect of a single relaxation training session on the cognitive performance of subjects suffering from a recurrent depressive disorder. Previous research primarily focused on cyclic relaxation activities [19,20,21]. The goal of the authors of this study was to test and examine whether generally available autogenic training of several minutes could be one way of independent practice with positive effects on cognitive function.

## 2. Material

Forty-two individuals aged 18–69 (*M* = 32.92, *SD* = 11.22; 27 females and 15 males) took part in the experiment. Twenty subjects in the study group had been diagnosed with recurrent depressive disorder (rDD group, F33) by a psychiatrist [22]. The sample size was estimated using the G*Power program [23].

The individuals in the rDD group were matched, taking into account the pharmacological treatment applied. Only the patients taking SSRI medications at the time of the study and who were receiving standard treatment with SSRIs were eligible to participate [24]. The subjects were recruited using the snowball method through social networks devoted to depression. Then, the mental state of the respondents was assessed by a psychiatrist.

The mean duration of pharmacotherapy at the time of the study was two weeks, and the severity of depressive symptoms as measured by the Beck Depression Inventory—Second Edition (BDI-II) scale was *M* = 18.81 (*SD* = 6.75), which indicated mild severity of depressive symptoms in the study group. Only the subjects who scored above 11 in the BDI-II test, indicating the presence of depressive symptoms, were included in the rDD group.

The comparison group comprised 22 healthy subjects (CG group), matched for age, gender, and education. The individuals for this group were recruited using the snowball method. The respondents also performed the BDI-II test to exclude people with symptoms of depression. In each case, the mental condition of the respondents was assessed by a psychiatrist.

There were no statistically significant differences in age, gender, or education between the groups analyzed. The social and demographic characteristics of the study group are presented in Table 1.

In the rDD group, subjects diagnosed with disorders other than axis disorders (rDD group, F33) were excluded from the study, together with subjects in both groups who had confirmed central nervous system damage, neurological diseases, neoplastic diseases, and other disease entities in the history, which could significantly affect their cognitive performance Only individuals who had had no previous experience with relaxation or mindfulness techniques were eligible to participate in the study (the relaxation training used in the study was the first experience of such kind for these individuals).

Participation in the study was free, and the subjects were recruited after they had given written informed consent to participate. Approval for the study was granted by the Bioethics Committee (No. RNN/328/18/KE).

## 3. Method

### 3.1. Evaluation of Attention Efficiency

The Attention and Perceptiveness Test (APT) by Ciechanowicz and Stańczak [25] was used to examine cognitive processes. APT belongs to the group of tests referred to as Continuous Performance Tests.

The test consists of four parallel versions: two versions with digits (6/9 and 3/8), one with letters, and one with graphic signs (geometric figures). Two parallel versions of the test were applied in this study: 6/9 and 3/8. TUS tests consist in drawing two symbols indicated in the instructions from a number of signs belonging to the same set (6/9 or 3/8). Within 3 min the test participant searches and crosses out from a sequence of digits only those given at the top of the page (digits depending on the version). After the test, the number of searched lines and the number of missed digits (6/9 or 3/8) are calculated. APT enables the following to be assessed:

*Perceptual Speed (PS)*—provides information on how quickly perceptual material is reviewed. The number of characters reviewed is important in this case. The higher the index, the higher the speed of perceptual work.

*Attention Fallibility (number of omissions, NO)*—provides information on the ability to perceive relevant stimuli. The signs that the individual should have crossed out but missed are of importance in this case. The higher the index, the greater the tendency to omit relevant stimuli [25].

### 3.2. Assessment of the Severity of Depressive Symptoms

Beck Depression Inventory—Second Edition (BDI-II) by Beck et al. was used to assess the dynamics of depressive disorder symptom severity (Polish adaptation by Łojek and Stańczak [25]. This questionnaire is a standardized tool to differentiate healthy individuals from those with depressive disorders and to assess the severity of depressive symptoms [26].

### 3.3. Relaxation Technique Used in the Study

Autogenic training developed by Johannes Heinrich Schultz was used as a relaxation method in this study [27]. This training was chosen because of its features, namely, it can be done alone, is universally accessible, and is fairly short (takes 15 min).

### 3.4. Study Procedure

Figure 1 shows the procedure of the study. It was applied to each of the subjects in the form presented below.

During the first stage of the study, each subject completed the BDI-II questionnaire and the 6/9 version of the APT test. Each subject was then asked to take a comfortable position on the couch. The investigator used a portable speaker to present the pre-prepared Schultz autogenic training, turned off the lights, and left the study room for the duration of the relaxation session. The training lasted 15 min. Following the relaxation training, the subjects performed the 3/8 version of the APT test.

### 3.5. Methods of Statistical Analysis

Selected descriptive methods and methods of statistical inference were applied to analyze the data. The first step was to use descriptive statistics for all quantitative parameters of the interpreted variables. The arithmetic mean (*M*) and standard deviation (*SD*) were calculated, and the symmetry of the distribution was verified. The normality of the distribution was checked using the Shapiro-Wilk test and the Lilliefors test. They allowed the hypothesis of normality of distribution to be rejected (*p* < 0.001). The Mann-Whitney U test was used to assess differences between independent variables, while the Wilcoxon signed-rank test was applied for dependent variables. Spearman’s rank correlation coefficient served to evaluate the correlations between the analyzed variables. In the conducted analyses, the adopted level of significance was *p* < 0.05 [28]. All statistical calculations were performed using STATISTICA PL software (version 13.1, StatSoft, Cracow, Poland).

## 4. Results

The mean score of the Perceptual Speed (PS) index before the introduction of relaxation training for the rDD group was *M* = 596 (*SD* = 61.12), while the number of omissions totaled *M* = 4.51 (*SD* = 3.22). For the CG group, these values were as follows: Perceptual Speed: *M* = 864 (*SD* = 87.31); number of omissions: *M* = 6.27 (*SD* = 4.67).

Following the relaxation training, the indexes specified above were:(a)rDD group: Perceptual Speed: *M* = 646.95 (*SD* = 94.62); number of omissions: *M* = 5.35 (*SD* = 3.03).(b)CG group: Perceptual Speed: *M* = 883.36 (*SD* = 91.59); number of omissions: *M* = 6.91 (*SD* = 5.29).

Table 2 presents a summary of the results obtained in the APT tests (6/9 and 3/8 versions) for the healthy subjects and the individuals diagnosed with recurrent depressive disorder. The Mann-Whitney U test for independent samples showed statistically significant differences in APT PS 6/9 and APT PS 3/8 scores between the study groups in reference to the Perceptual Speed index. These differences are significant and evident both before and after the introduction of relaxation training (in favor of the comparison group).

In the next step of the analysis, the performance level of both parts of the APT test before and after the relaxation technique was compared in each study group. Table 3 shows the results obtained for the entire study group and with a breakdown between the subjects in the recurrent depressive disorder group (rDD) and the comparison group (CG).

A statistically significant difference in the index value for ‘Perceptual Speed’ before and after applied relaxation training was observed. These differences were found for those with depression and for the entire study group (*N* = 42). In each case, the relaxation training increased Perceptual Speed.

In the case of the ‘number of omissions’ index, no statistically significant differences were found for any of the analyzed groups, but as presented in Table 3, the number of omissions after relaxation training decreased in each of the studied groups.

Spearman’s rank correlation analysis revealed associations between the level of performance on the Perceptual Speed index of the APT test and the severity of depressive symptoms in the study group with rDD:Before relaxation training (6/9 APT version): *r* = −0.864 (*p* < 0.001);After relaxation training (3/8 APT version): *r* = −0.785 (*p* < 0.001).

The results obtained indicate a significant association of illness symptom severity with reduced performance in the group with depressive disorder symptoms.

## 5. Discussion

This study indicates that relaxation training has a positive effect on attention efficiency. The results obtained make it possible to conclude that even a 15-min-long one-time relaxation exercise has a beneficial impact on the efficiency of attention in people with depression.

There are a number of techniques designed to regenerate and relax the body and mind as well as improve human functioning. Most of them, including autogenic training by Schultz, develop mindfulness, which refers to a specific type of attention, i.e., conscious, moment-directed, and non-judgmental [29]. It allows everyone to become a whole self in the face of experiences coming both from within (thoughts or feelings) and from the surroundings (such as other people’s behavior). Mindfulness practice is based on teaching participants to pay full attention to their bodies and the outside world [30]. Mindfulness can be developed through meditation or relaxation techniques [24]. Because of their properties, relaxation techniques are used in the treatment of numerous mental [31] and somatic disorders [32]. The effectiveness of such programs is well documented in terms of regulating emotional responses in depression (relapse prevention), anxiety disorders, obsessive-compulsive disorders, addictions, and eating disorders [33]. Moreover, converging evidence supports the hypothesis that mindfulness practice improves cognitive abilities, particularly the ability to maintain attention and think more flexibly [34]. Developing mindfulness has a positive effect on brain function and reduces the likelihood of symptoms of anxiety and depression.

Neuroimaging studies indicate favorable changes in the gray matter area, increased cortical thickness in the prefrontal cortex (PFC) [35], and an increase in serotonin (5-HT) levels in the blood [36]. Brief relaxation practice led to significant frontal lobe activation during arithmetic tasks [37].

Mindfulness training using smartphone apps is an effective intervention for reducing perinatal depression in women in early pregnancy who are potentially at risk for depression [38].

Mindfulness principles have been incorporated into other prominent therapeutic interventions such as dialectical behavior therapy (DBT) and acceptance and commitment therapy (ACT). In addition, mindfulness is increasingly being studied in the context of cognitive behavior therapy (CBT) for individuals with emotional disorders [39].

Gangadharan et al. [40] attempted to assess the prevalence of depressive symptoms, anxiety, and levels of perceived stress among nursing students (*N* = 218) who received progressive muscle relaxation techniques. It was found that the frequency of the assessed indicators was significantly (*p* = 0.0001) reduced as a consequence of the applied intervention: 75% (post-intervention—34.4%) for depressive symptoms; 119% (post-intervention—54.6%) for anxiety symptoms; and 78% (post-intervention—35.8%) for levels of perceived stress, respectively [40].

One of the significant symptoms of depressive disorders is a sense of diminished intellectual performance. Patients often report decreased mental dynamics, as well as difficulty focusing attention and learning new information [41]. Anxiety—as a component of depressive disorders—interferes with the functioning of the visuospatial sketchpad, which is part of working memory [42].

A meta-analysis of cognitive performance in depressed individuals by McDermott & Ebmeier [43] revealed significant correlations between depression severity and cognitive performance in episodic memory, information processing speed, and executive function [43]. The study showed that 63.3% (including 65% of females and 60% of males) with diagnosed depression had cognitive impairment (the rate in healthy individuals was 3.3%, *p* < 0.001). One of the most common cognitive impairments in patients with depression is decreased efficiency of attention and memory processes [44]. A study conducted by Lee et al. [1] also supports the cited data. It was shown that even during the first episode of major depression, subjects (*N* = 644) exhibited significant cognitive deficits in psychomotor speed, learning, visual memory, and attention [1].

The study described herein confirms scientific reports presented by other researchers on the prevalence of cognitive dysfunction in people with depressive disorders, as well as the association of the severity of illness symptoms with reduced cognitive performance [17,45]. Perceptual speed is significantly weaker in depressed individuals compared to healthy individuals. This phenomenon can be linked to difficulties in focusing attention with a reduced ability to effectively search the perceptual field and perceptual efficiency [46].

## 6. Practical Implications of the Study

Relaxation is one form of support in treating depression that can co-exist with pharmacotherapy and psychotherapy. It has a positive effect on mood, as well as cognitive processes. Even a single practice based on autogenic training by Schultz leads to improvements in attention and perceptiveness, which may suggest that systematic relaxation will significantly affect the functioning of people with depression.

Practicing mindfulness has a number of benefits within cognitive processes. Even a few minutes of practice increases perceptual speed, hence attention concentration also improves. People with depression suffer from cognitive dysfunction. Incorporating relaxation training into treatment can reduce this problem.

The study presented herein confirms previous reports of the positive effects of using mindfulness to treat depressive disorders. The collected results may contribute to the development of alternative methods of depression therapy that can be used regardless of the age of the patients. It is also important that relaxation techniques can be used independently by patients at a convenient time for them. It is useful to combine these techniques with pharmacotherapy and psychotherapy. It is worth encouraging doctors of various specialties to promote the described method of symptom relief.

## 7. Conclusions

The continuous performance test (APT) is a reliable indicator of impaired attention efficiency among patients with depressive symptoms compared to healthy subjects.

Fifteen-min-long one-time relaxation exercise has a beneficial effect on attention efficiency measured by APT in people with depression.

## 8. Limitations

Limitations of the study may include the size of the study sample and the inclusion of individuals with mild severity of illness symptoms in the group of subjects with recurrent depressive disorder. The authors of this paper are aware of these limitations. However, the statistical analyses conducted revealed significant relationships between the variables analyzed.The study presented here does not provide direct evidence (e.g., in the form of imaging findings) to explain the mechanisms underlying the association between 15-min-long one-time relaxation exercise and attention deficits in depression. However, it allows for the formulation of hypotheses to explain the relationships obtained.The size of the group of depressed individuals does not allow for additional analyses that would indicate the connections between attention deficits and different severity of depressive symptoms or different clinical courses of depression.

## Figures and Tables

**Figure 1 ijerph-19-06473-f001:**
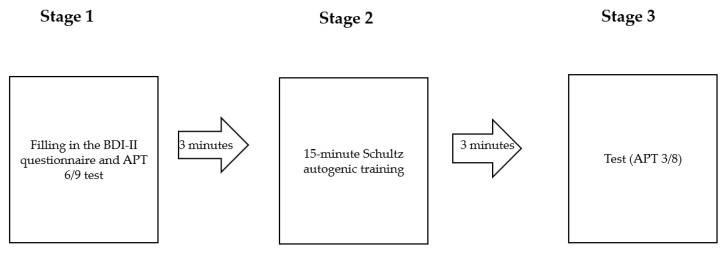
Study procedure. BDI-II—Beck Depression Inventory—Second Edition; APT 6/9—Attention and Perceptiveness Test, 6/9 version; APT 3/8—Attention and Perceptiveness Test, 3/8 version.

**Table 1 ijerph-19-06473-t001:** The social and demographic characteristics of the study group.

Variables		rDD *N* = 20		CG *N* = 22		All *N* = 42	
		*M*	*SD*	*M*	*SD*	*M*	*SD*
**Age (Years)**		**35.91**	**14.09**	**30.23**	**7.05**	**32.92**	**11.22**
Severity of depressive disorder symptoms as measured by BDI-II		18.81	6.75	-		-	
Gender	Females	* **N** *	* **%** *	* **N** *	* **%** *	* **N** *	* **%** *
		15	75.01	12	45.45	27	64.29
	Males	5	25.01	10	54.54	15	35.71
Education	Higher	12	60.01	15	68.18	27	64.29
	Secondary	8	40.01	7	31.82	15	35.71
	Rural settlement/ Town < 50,000 inhabitants	7	35.01	5	22.73	12	28.57
	City > 50,000 inhabitants	13	65.01	17	77.27	30	71.43

rDD—recurrent depressive disorder; CG—comparison group; All—entire study group; *N*—size; *M*—mean; *SD*—standard deviation; %—percentage; BDI-II—Beck Depression Inventory—Second Edition.

**Table 2 ijerph-19-06473-t002:** Comparison of the results of the two versions of the APT test in the study groups (*N* = 42).

Variables		rDD (*N* = 20) Sum of Ranks	CG (*N* = 22) Sum of Ranks	Mann-Whitney U Test	
				*Z*	*p*
**Before** **relaxation training**	PS 6/9	**210.51**	692.51	5.515	<0.001 *
	NO 6/9	386	517	1.096	0.273
**After** **relaxation training**	PS 3/8	230.5	672.5	5.012	<0.001 *
	NO 3/8	387.5	515.5	1.058	0.291

PS 6/9—Perceptual Speed for APT 6/9 version; NO 6/9—Number of omission for APT 6/9 version; PS 3/8—Perceptual Speed for APT version; NO 3/8—Number of omission for APT 3/8 version; rDD—Subjects with recurrent depressive disorder; CG—Comparison group; *—*p* statistically significant.

**Table 3 ijerph-19-06473-t003:** Comparison of the level of performance of the two parts of the APT test before and after relaxation training.

**Variables**	**rDD** **(*N* = 20)**		**Wilcoxon** **Signed-Rank Test**	
	* **M** *	* **SD** *	* **Z** *	* **p** *
PS 6/9 Before relaxation training	596.01	61.12	2.87	0.004 *
PS 3/8 After relaxation training	646.95	94.62		
NO 6/9 Before relaxation training	5.35	3.47	0.118	0.915
NO 3/8 After relaxation training	4.51	3.22		
**Variables**	**CG** **(*N* = 22)**		**Wilcoxon** **Signed-Rank Test**	
	* **M** *	* **SD** *	* **Z** *	* **p** *
PS 6/9 Before relaxation training	864.1	87.31	0.672	0.501
PS 3/9 After relaxation training	883.3	91.59		
NO 6/9 Before relaxation training	6.91	5.29	0.181	0.856
NO 3/8 After relaxation training	6.27	4.67		

PS 6/9—Perceptual Speed for APT 6/9 version; NO 6/9—Number of omission for APT 6/9 version; PS 3/8—Perceptual Speed for APT version; NO 3/8—Number of omission for APT 3/8 version; rDD—Subjects with recurrent depressive disorder; CG—Comparison group; *—*p* statistically significant.

## Data Availability

Not applicable.

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
