# Peer review of "The Impact of One-Time Relaxation Training on Attention Efficiency Measured by Continuous Performance Test in Depressive Disorders"

_ijerph, 2022, doi:10.3390/ijerph19116473_

Round 1

Reviewer 1 Report

・15 line:  No need to state the number of people who are eligible twice. Please remove the parentheses.

・29-32 line: Conclusions should be in writing, not in bullet points.

・13 and 31 line: I don't think the descriptions are consistent, such as a brief (15-minute-long) relaxation training and a brief (15-minute-long) one-time relaxation exercise, are they the same thing? Or are they different?

・How was the sample size for this subject determined?

・Table 1:The number of people in the CG group is indicated as 20, which is correct, the one in the text (22) or the one in the table? Also, as for the BDI-2 scores, shouldn't the ALL portion of the BDI-2 be the rDD group scores as they are?

・Is there any influence of covariates? Have you considered that consideration?

Author Response

Thank you for preparing the review. The introduced changes are marked in the text.

Regarding the following comments:

Ad. 1.

“15 line:  No need to state the number of people who are eligible twice. Please remove the parentheses”.

Ad. 2.

29-32 line: Conclusions should be in writing, not in bullet points”.

Ad 3.

“13 and 31 line: I don't think the descriptions are consistent, such as a brief (15-minute-long) relaxation training and a brief (15-minute-long) one-time relaxation exercise, are they the same thing? Or are they different?”.

The above comments of the Reviewer were taken into account.

Ad 4.

How was the sample size for this subject determined?”

The sample size was estimated using the G*Power program (line 93-94, 340-341).

Faul F, Erdfelder E, Lang AG, Buchner A. G*Power 3: a flexible statistical power analysis program for the social, behavioral, and biomedical sciences. Behav Res Methods. 2007;39(2):175-91. doi: 10.3758/bf03193146.

Ad. 5.

Table 1The number of people in the CG group is indicated as 20, which is correct, the one in the text (22) or the one in the table? Also, as for the BDI-2 scores, shouldn't the ALL portion of the BDI-2 be the rDD group scores as they are?”

22 healthy people participated in the study. By mistake, we entered the value 20 in the table 1. We made a correction.

Only the subjects who scored above 11 in the BDI-II test, indicating the presence of de-pressive symptoms, were included in the rDD group.

The text provides the mean value of the BDI-II test only for these people.

Ad 6.

Is there any influence of covariates? Have you considered that consideration?”

Thank you for this valuable attention.

When formulating the inclusion and exclusion criteria, we tried to minimize the impact of other independent variables.

Reviewer 2 Report

I appreciate the opportunity to review this manuscript and would like to congratulate the authors on a job well done. I believe that the authors have carried out a study of high scientific and academic interest, which provides important information. I must say that I find it a well written work that presents important and valuable data.

The Introduction section has been written to provide sufficient information for readers to understand the research hypotheses and the importance of conducting studies along these lines, and the objectives are well defined in the Introduction.

Methodology: this section explains how the study was carried out and details the research design and measures used.

All the interpretations that the authors have made of the data obtained are coherent.

Literature cited: The literature cited is relevant to the study.

Significance and novelty: As it stands, the results are novel and important enough for this journal.

The author/s have made a good job, although I contribute here some suggestions for the improvement of the quality of the document:

  • Explain more the possible practical applications of the study carried out.

Author Response

Thank you for preparing the review. The introduced changes are marked in the text.

Regarding the following comments:

Ad. 1.

“Explain more the possible practical applications of the study carried out”.

We supplemented the discussion with practical implications (line 301-306).

Reviewer 3 Report

The authors present a study comparing individuals with recurrent depressive disorder with healthy controls on cognitive functioning before and after a brief autogenic relaxation session. The results are promising and have implications for treatment of recurrent depressive disorder (and mild depressive disorders as well). There are a few issues I would like the authors to address:

  • I advise the authors to start the introduction with a description of (the cognitive deficits/problems) of depressive disorders. At the moment, it is quite unclear until the final paragraphs that the proposed intervention/relaxation session will be aimed at improving cognitive functioning of individuals with (recurrent) depressive disorders.
  • The opening sentences in particular put the reader on the wrong track due to the mention of 'altered state of consciousness' (which I think is not a symptom of depression?).
  • The authors should emphasize or elaborate on the importance of e.g. perceptual speed/attention/omission of relevant stimuli or other cognitive functions for individuals with depressive disorder - and, conversely, detail how impaired perceptual speed/attention is symptomatic of depression and/or affects individuals' wellbeing (or exacerbates the disorder). This is important especially since the main symptoms of depression are depressed mood, anhedonia, rumination, self-esteem issues etc. etc.
  • The methodology, and in particular the utilized dependent measure (the APT) should be described in more detail. It is unclear what this task or measure consists of, and how to interpret the means provided in the results section. Is this a reaction times task? What is the difference between the 6/9 and 3/8 versions? How many trials do participants have to complete and how is the final score calculated?
  • Similarly, there is no information about the autogenic relaxation task (p.4) - what does this consist of? guided meditation? Breathing exercises? Where can readers find this training (youtube?)?
  • Related - can the increase in scores after the relaxation training be a result of learning? If these tasks are very comparable in nature, the fact that participants completed one could have positively affected the second time they did a similar/same task? This is a concern especially because the time between the 2 tasks is very brief.
  • Should the scores on the APT 6/9 and APT 3/8 not be standardized before comparisons can be statistically made? Have such comparisons been made in previous studies as well?
  • I don't think it is necessary to report the 'overall' results since the explicit aim is to compare the 2 groups (Table 3). What I miss - but perhaps I am misinterpreting the Table(s) - is a 2x2 analysis (within/between) analysis for both PS and NO.
  • Did you also administer the Beck inventory to the healthy controls? To make sure they were indeed scoring (very) low on depressive symptoms and thus to verify they were indeed healthy controls?
  • Where exactly were the participants in the rDD group recruited? in a hospital, via a therapist/psychiatrist? More information is needed here.
  • Why are only correlations between depressive symptoms and PS reported, but not between depressive symptoms and NO?
  • Did the authors control for participants' age & gender in their analyses? 
  • Discussion is overall written in the same manner as the introduction, i.e. presenting the cognitive consequences of depressive disorder quite late (mid page 7). Further, the limitations & suggestions section needs to be expanded (see the comments above) and the conclusion should reflect the entire study not just the APT, and should be the last paragraph of the discussion section.

Author Response

Thank you for preparing the review. The introduced changes are marked in the text.

Regarding the following comments:

Ad 1.

“I advise the authors to start the introduction with a description of (the cognitive deficits/problems) of depressive disorders. At the moment, it is quite unclear until the final paragraphs that the proposed intervention/relaxation session will be aimed at improving cognitive functioning of individuals with (recurrent) depressive disorders”.

Ad 2.

“The authors should emphasize or elaborate on the importance of e.g. perceptual speed/attention/omission of relevant stimuli or other cognitive functions for individuals with depressive disorder - and, conversely, detail how impaired perceptual speed/attention is symptomatic of depression and/or affects individuals' wellbeing (or exacerbates the disorder). This is important especially since the main symptoms of depression are depressed mood, anhedonia, rumination, self-esteem issues etc.”.

Cognitive impairment in depression is well understood and described. I have researched this issue many times in my earlier works (below I present some of them). In this article, I wanted to focus mainly on the benefits from a short relaxation training.

However, as suggested by the Reviewer, I have added a few items to the references section.

Interleukin 1 level, cognitive performance, and severity of depressive symptoms in patients treated with systemic anticancer therapy: a prospective study.

Jasionowska J, Talarowska M, Kalinka E, Skiba A, Szemraj J, Mikołajczyk I, Gałecki P.Croat Med J. 2019 Apr 30;60(2):166-173. doi: 10.3325/cmj.2019.60.166.PMID: 31044590 Free PMC article.

Inflammatory theory of depression.

Gałecki P, Talarowska M.Psychiatr Pol. 2018 Jun 30;52(3):437-447. doi: 10.12740/PP/76863. Epub 2018 Jun 30.PMID: 30218560 Free article. Review. English, Polish.

Expression levels of interferon-ɣ and type 2 deiodinase in patients diagnosed with recurrent depressive disorders.

Gałecka E, Talarowska M, Maes M, Su KP, Górski P, Kumor-Kisielewska A, Szemraj J.Pharmacol Rep. 2018 Feb;70(1):133-138. doi: 10.1016/j.pharep.2017.08.009. Epub 2018 Feb 4.PMID: 29367100

Is there a link between TNF gene expression and cognitive deficits in depression?

Bobińska K, Gałecka E, Szemraj J, Gałecki P, Talarowska M.Acta Biochim Pol. 2017;64(1):65-73. doi: 10.18388/abp.2016_1276. Epub 2016 Dec 16.PMID: 27991935 Free article.

Polymorphisms of iodothyronine deiodinases (DIO1, DIO3) genes are not associated with recurrent depressive disorder.

Gałecka E, Talarowska M, Maes M, Su KP, Górski P, Szemraj J.Pharmacol Rep. 2016 Oct;68(5):913-7. doi: 10.1016/j.pharep.2016.04.019. Epub 2016 May 12.PMID: 27351946

Polymorphisms in the type I deiodinase gene and frontal function in recurrent depressive disorder.

Gałecka E, Talarowska M, Orzechowska A, Górski P, Szemraj J.Adv Med Sci. 2016 Sep;61(2):198-202. doi: 10.1016/j.advms.2015.12.008. Epub 2016 Jan 13.PMID: 26866568

The role of MMP genes in recurrent depressive disorders and cognitive functions.

Bobińska K, Szemraj J, Gałecki P, Talarowska M.Acta Neuropsychiatr. 2016 Aug;28(4):221-31. doi: 10.1017/neu.2015.72. Epub 2016 Feb 9.PMID: 26856768

Serum KIBRA mRNA and Protein Expression and Cognitive Functions in Depression.

Talarowska M, Szemraj J, Kowalczyk M, Gałecki P.Med Sci Monit. 2016 Jan 15;22:152-60. doi: 10.12659/msm.895200.PMID: 26768155 Free PMC article.

Cognition and Emotions in Recurrent Depressive Disorders - The Role of Inflammation and the Kynurenine Pathway.

Talarowska M, Galecki P.Curr Pharm Des. 2016;22(8):955-62. doi: 10.2174/1381612822666151230110738.PMID: 26714730 Review.

Autobiographical memory dysfunctions in depressive disorders.

Talarowska M, Berk M, Maes M, Gałecki P.Psychiatry Clin Neurosci. 2016 Feb;70(2):100-8. doi: 10.1111/pcn.12370. Epub 2015 Nov 29.PMID: 26522618 Free article. Review.

[Emotional and language prosody and working memory in patients with depression].

Talarowska M, Zajączkowska M, Gałecki P.Pol Merkur Lekarski. 2015 May;38(227):269-72.PMID: 26039021 Clinical Trial. Polish.

Mechanisms underlying neurocognitive dysfunctions in recurrent major depression.

Gałecki P, Talarowska M, Anderson G, Berk M, Maes M.Med Sci Monit. 2015 May 27;21:1535-47. doi: 10.12659/MSM.893176.PMID: 26017336 Free PMC article. Review.

Cognitive functions in first-episode depression and recurrent depressive disorder.

Talarowska M, Zajączkowska M, Gałecki P.Psychiatr Danub. 2015 Mar;27(1):38-43.PMID: 25751430

Manganese superoxide dismutase gene expression and cognitive functions in recurrent depressive disorder.

Talarowska M, Orzechowska A, Szemraj J, Su KP, Maes M, Gałecki P.Neuropsychobiology. 2014;70(1):23-8. doi: 10.1159/000363340. Epub 2014 Aug 21.PMID: 25171019

Myeloperoxidase gene expression and cognitive functions in depression.

Talarowska M, Szemraj J, Gałecki P.Adv Med Sci. 2015 Mar;60(1):1-5. doi: 10.1016/j.advms.2014.06.001. Epub 2014 Jun 26.PMID: 25038328

ASMT gene expression correlates with cognitive impairment in patients with recurrent depressive disorder.

Talarowska M, Szemraj J, Zajączkowska M, Gałecki P.Med Sci Monit. 2014 Jun 2;20:905-12. doi: 10.12659/MSM.890160.PMID: 24881886 Free PMC article.

Thiol protein groups correlate with cognitive impairment in patients with recurrent depressive disorder.

Gałecki P, Talarowska M, Bobińska K, Kowalczyk E, Gałecka E, Lewiński A.Neuro Endocrinol Lett. 2013;34(8):780-6.PMID: 24522026

Impact of oxidative/nitrosative stress and inflammation on cognitive functions in patients with recurrent depressive disorders.

Talarowska M, Bobińska K, Zajączkowska M, Su KP, Maes M, Gałecki P.Med Sci Monit. 2014 Jan 24;20:110-5. doi: 10.12659/MSM.889853.PMID: 24457625 Free PMC article.

COX-2 gene expression is correlated with cognitive function in recurrent depressive disorder.

Gałecki P, Talarowska M, Bobińska K, Szemraj J.Psychiatry Res. 2014 Feb 28;215(2):488-90. doi: 10.1016/j.psychres.2013.12.017. Epub 2013 Dec 18.PMID: 24388097

Working memory impairment as a common component in recurrent depressive disorder and certain somatic diseases.

Galecki P, Talarowska M, Moczulski D, Bobinska K, Opuchlik K, Galecka E, Florkowski A, Lewinski A.Neuro Endocrinol Lett. 2013;34(5):436-45.PMID: 23922050

[Correlations between working memory effectiveness and depression levels after pharmacological therapy].

Talarowska M, Zboralski K, Gałecki P.Psychiatr Pol. 2013 Mar-Apr;47(2):255-67.PMID: 23888759 Polish.

[Are there any differences in the working memory of men and women?].

Talarowska M, Florkowski A, Chamielec M, Gałecki P.Pol Merkur Lekarski. 2013 Jan;34(199):29-32.PMID: 23488281 Polish.

[Results of the Trail Making Test among patients suffering from depressive disorders and organic depressive disorders].

Talarowska M, Krzysztof Z, Mossakowska-Wójcik J, Gałecki P.Psychiatr Pol. 2012 Mar-Apr;46(2):273-82.PMID: 23214397 Polish.

Total antioxidant status correlates with cognitive impairment in patients with recurrent depressive disorder.

Talarowska M, Gałecki P, Maes M, Bobińska K, Kowalczyk E.Neurochem Res. 2012 Aug;37(8):1761-7. doi: 10.1007/s11064-012-0788-z. Epub 2012 May 5.PMID: 22562440

Nitric oxide plasma concentration associated with cognitive impairment in patients with recurrent depressive disorder.

Talarowska M, Gałecki P, Maes M, Orzechowska A, Chamielec M, Bartosz G, Kowalczyk E.Neurosci Lett. 2012 Feb 29;510(2):127-31. doi: 10.1016/j.neulet.2012.01.018. Epub 2012 Jan 17.PMID: 22273980

[The role of the right hemisphere in the aetiology of the depressive disorders].

Talarowska M, Orzechowska A, Zboralski K, Gałecki P.Psychiatr Pol. 2011 Jul-Aug;45(4):563-72.PMID: 22232982 Review. Polish.

[Results of the Benton Visual Retention Test and the Bender Visual--Motor Gestalt Test among patients suffer from depressive disorders and organic depressive disorders].

Talarowska M, Florkowski A, Zboralski K, Gałecki P.Psychiatr Pol. 2011 Jul-Aug;45(4):495-504.PMID: 22232976 Polish.

Malondialdehyde plasma concentration correlates with declarative and working memory in patients with recurrent depressive disorder.

Talarowska M, Gałecki P, Maes M, Gardner A, Chamielec M, Orzechowska A, Bobińska K, Kowalczyk E.Mol Biol Rep. 2012 May;39(5):5359-66. doi: 10.1007/s11033-011-1335-8. Epub 2011 Dec 15.PMID: 22170602

Auditory-verbal declarative and operating memory among patients suffering from depressive disorders - preliminary study.

Talarowska M, Florkowski A, Zboralski K, Berent D, Wierzbiński P, Gałecki P.Adv Med Sci. 2010;55(2):317-27. doi: 10.2478/v10039-010-0053-0.PMID: 21163755 Clinical Trial.

[Differences in cognitive functioning of men and women with a diagnosis of depression].

Fitas A, Berent D, Talarowska M.Pol Merkur Lekarski. 2010 Mar;28(165):199-202.PMID: 20815167 Polish.

[Cognitive functions and depression].

Talarowska M, Florkowski A, Gałecki P, Wysokiński A, Zboralski K.Psychiatr Pol. 2009 Jan-Feb;43(1):31-40.PMID: 19694398 Review. Polish.

Ad 3.

“The opening sentences in particular put the reader on the wrong track due to the mention of 'altered state of consciousness' (which I think is not a symptom of depression?)”.

The introduction has been changed in line with the Reviewer's suggestion.

Ad 4.

“The methodology, and in particular the utilized dependent measure (the APT) should be described in more detail. It is unclear what this task or measure consists of, and how to interpret the means provided in the results section. Is this a reaction times task? What is the difference between the 6/9 and 3/8 versions? How many trials do participants have to complete and how is the final score calculated?”.

Ad 5.

“Related - can the increase in scores after the relaxation training be a result of learning? If these tasks are very comparable in nature, the fact that participants completed one could have positively affected the second time they did a similar/same task? This is a concern especially because the time between the 2 tasks is very brief”.

We have added the indicated information (line 134-138).

Ad 6.

“Similarly, there is no information about the autogenic relaxation task (p.4) - what does this consist of? guided meditation? Breathing exercises? Where can readers find this training (youtube?)?”.

Autogenic training is a form of guided relaxation. The participant follows simple instructions. The tasks consist of focusing on particular parts of the body, e.g. the right hand, the left leg. The relaxation is available for free on youtube.

Ad 7.

“Should the scores on the APT 6/9 and APT 3/8 not be standardized before comparisons can be statistically made? Have such comparisons been made in previous studies as well?”.

We used raw results in the statistical analysis.

Ad 8.

“I don't think it is necessary to report the 'overall' results since the explicit aim is to compare the 2 groups (Table 3). What I miss - but perhaps I am misinterpreting the Table(s) - is a 2x2 analysis (within/between) analysis for both PS and NO”.

Table 3 was changed.

Ad 9.

“Did you also administer the Beck inventory to the healthy controls? To make sure they were indeed scoring (very) low on depressive symptoms and thus to verify they were indeed healthy controls?”.

The respondents also performed BDI-II to exclude people with symptoms of depression (lin 107-109).

Ad 10.

“Where exactly were the participants in the rDD group recruited? in a hospital, via a therapist/psychiatrist? More information is needed here”.

Descriptions of the studied groups were supplemented.

Ad 11.

“Why are only correlations between depressive symptoms and PS reported, but not between depressive symptoms and NO?”.

In case of the ‘number of omissions’ index, no statistically significant differences were found for any of the analyzed groups.

Ad 12.

“Did the authors control for participants' age & gender in their analyses?”

Yes (line 107-113).

Ad 13.

“Discussion is overall written in the same manner as the introduction, i.e. presenting the cognitive consequences of depressive disorder quite late (mid page 7). Further, the limitations & suggestions section needs to be expanded (see the comments above) and the conclusion should reflect the entire study not just the APT, and should be the last paragraph of the discussion section”.

The conclusions section and the limitation were changed.

Round 2

Reviewer 1 Report

Thank you for responding to my comments.

Author Response

Thank you. 

Reviewer 3 Report

Although some parts have been revised by the authors, my overall comments remain the same (see report 1); I think the authors did not address my main concerns at all in this version of the manuscript. I fully understand the author(s) is/are an expert on depression & cognitive deficits - and I am fully aware of the link that exists - but my point was that a reader may not be, hence this link should be elaborated upon. Simply adding references does not suffice.

Other important comments have also not been addressed - e.g. the limitation section, the way the final scores of the tests are computed. The methods section is still lacking clarity (and as a result, so do the results).
